# Degradation Behavior of Arc-Sprayed Zinc Aluminum Alloy Coatings for the Vessel Yongle in the South China Sea

**Guo-Sheng Huang** [1,*] **, Zi-Lin Li** [2] **, Xiao-Shuo Zhao** [3] **, Yong-Lei Xin** [1] **, Li Ma** [1] **, Ming-Xian Sun** [1] **and Xiang-Bo Li** [1]

1    State Key Laboratory for Marine Corrosion and Protection, Luoyang Ship Material Research Institute, Qingdao 266237, China
2    702 Institution of China Shipbuilding Industry Group Co., Ltd., Wuxi 214082, China
3    State Key Laboratory of Advanced Technology for Materials Synthesis and Processing, Wuhan University of Technology, Wuhan 430070, China
*    Correspondence: huanggs@sunrui.net

**Abstract:** Since thermally sprayed zinc and aluminum coatings were invented 100 years ago, they have realized extensive industrial applications for steel structure protection in a variety of fields for nearly 100 years and have been proven to be effective and reliable. However, it has seldom been reported in the ship industry in China since many workers worry about the risk of rapid corrosion, especially in harsh environments such as the South China Sea. In this paper, three kinds of arc-sprayed zinc aluminum coatings were tested to choose the best coating system for application on the research vessel Yongle by electrochemical behavior and a long-term atmospheric exposure experiment. The variation of the corrosion rate and the bonding strength was used to clarify the long-term protection performance. The results show that Zn15Al has the lowest corrosion ($R_p$ larger than 2200 $\Omega \cdot cm^2$) among the three kinds of coatings and has a bonding strength larger than 6.38 MPa after a 5 year test. The performance of the coatings in the South China Sea indicates that they can provide excellent protection for the hull above the waterline of the Yongle vessel in the 3 year test. It could be predicted that thermally sprayed zinc aluminum coating has vast application potential in the South China Sea due to its excellent anticorrosion performance.

**Keywords:** zinc aluminum coating; arc spraying; the research vessel Yongle; South China Sea

## 1. Introduction

Steel structures suffer from more serious corrosion problems in the South China Sea than on islands due to high humidity, high temperature, high salinity, and strengthened solar irradiance [1–4]. Organic coatings, as an effective method for corrosion protection, experience a higher aging rate than those in the Yellow Sea and the East China Sea, which leads to a high corrosion risk for long-term protection. Thermally sprayed aluminum [5,6], zinc [7,8] and aluminum-zinc alloy [6,9,10] coatings can provide long term protection for steel structures in most types of environments, which are rarely reported in the South China Sea. According to some reports, zinc has the same corrosion rate as steel in most marine environments [11,12], so many people worry that thermally sprayed coatings cannot provide enough long-term protection for steel in some harsh situations [13–15]. There has been much research on this topic, which has mainly focused on the corrosion rate of zinc and its alloy coatings. Y. Li reported that the addition of Al to Zn increases the corrosion resistance properties of galvanized coating after two years of exposure in a seawater environment due to its optimum combination that is resistive to uniform and pitting corrosion [16]. Gulec et al. [17] examined the effects of Al addition in Zn coatings on the corrosion characteristics of steel exposed to an accelerated condition and indicated the pronounced corrosion resistance of the Zn/15Al coating [17]. In the Al-Zn coating, Zn provides cathodic protection while Al provides erosion resistance [18,19]. A schematic was proposed by Lee H. that explains the corrosion process of Al-Zn pseudo-alloy coating

in 3.5 wt.% NaCl solution from the deposition of coating and initiation of corrosion to longer exposure durations [20]. Some research was focused on the passivation film and filling the pores, which can elongate the protection duration by decreasing the corrosion rate [21,22]. It can be seen that the investigation into the failure behavior of the zinc and its alloy coatings was mainly focused on the corrosion rate. Additionally, the results also showed that the zinc aluminum coatings can bear much longer-term consumption in the marine environment.

However, the variation in mechanical properties during the service of the thermally sprayed zinc and its alloy coatings has not been well investigated. As we know, the failure of a metal or organic coating is determined by complicated factors, not only the corrosion rate. Peeling off, induced by stress, is the main failure pattern of coatings. In some conditions, metal coatings peel off on a large scale because of residual stress when they are being constructed if the coating quality is not well controlled. Corrosion products accumulate in the coating or at the interface and can generate a very high level of stress, which could be higher than the adhesion and cohesion of the coatings on the substrate. The theory was that strain on the coating caused by the corrosion products, together with internal stress in the coating, gave stress levels above the cohesive strength of the coating. In the cracks, bare steel is exposed, resulting in enhanced corrosion and new cracking. In this way, the degradation of the coating propagated rapidly [23]. Furthermore, zinc and its alloy coatings are sacrificial coatings; corrosion is inevitable in most circumstances. It is an important factor that cannot be neglected for zinc and its alloy coatings.

In this paper, the degradation behavior of the arc-sprayed zinc and its alloy coatings was investigated by taking time-dependent corrosion rates and bonding strength variations into consideration. The corrosion behavior and failure mechanisms were discussed to determine the long-term protection performances of zinc alloy coatings by long-term atmospheric exposure and an electrochemical test.

## 2. Materials and Methods

### 2.1. Raw Materials and Substrate

The substrate of the test sample was Q235 steel purchased from Shanghai BaoSteel Co., Ltd. Shanghai, China (nominal composites as follows: C 0.14–0.22 wt.%, Mn 0.30–0.65 wt.%, Si $\leq$ 0.30 wt.%, S $\leq$ 0.050 wt.%, and P $\leq$ 0.045 wt.%), sandblasted for 15 min under the pressure of 0.6 MPa of compressed air before spraying. The equipment (L-arc 400) for spraying samples and field construction was purchased from United Coatings Technology Co., Ltd. (Beijing, China). The parameters for spraying zinc aluminum coating were: 0.5 MPa atomic pressure for compressed dry air, a standoff distance of 200–300 mm from the gun exit to the substrate, perpendicular to the substrate, and a wire feeding rate of 5 cm/s. The zinc and zinc alloy wire had a diameter of 3 mm, and the purity was higher than 99.5 wt.%. The zinc aluminum alloy types included Zn, Zn15Al, and Zn45Al, which were purchased from Shijiazhuang Xinri Zinc Company, Shijiazhuang, China.

### 2.2. Test Methods for Performances

The coating thickness data were collected by an Elcometer 456 Coating Thickness Gauge (Shenzhen, China). For samples, the thickness was measured randomly on the surface at five points, and the average thickness was calculated for each sample. For the ship's hull during the coating construction, the thickness was measured regularly on a plane surface within a 10 cm × 10 cm area, and the average thickness was calculated for each sampling location. The bonding strength (BS) of the coatings was collected by an Elcometer model 108 test machine (Elcometer, London, UK). For samples, the BS was measured for 3–5 parallel samples. For the ship's hull during the coating construction, the BS was measured regularly on a plane surface at one location per 50 m$^2$.

Samples for surface and cross-section morphology observation were cut from the sprayed samples into 10 mm × 10 mm small pieces by electrical discharge wire cutting. Then, the samples for the cross-sectional test were sealed using a hot mounting machine,

and only the cross-sections were left for examination. The cross-sections were abraded with 100#, 400#, 800#, 1200#, and 2000# sandpaper in sequence. Afterwards, the coatings were cleaned with water, wiped with alcohol, and dried with an air dryer. The samples from the surface test were left in sprayed status. Finally, the micro-morphologies of the coatings were observed with a SEM. Field emission-scanning electron microscopy (FE-SEM) images were obtained by a Hitachi SU8020 FE-SEM microscope (Tokyo, Japan). Energy dispersive spectrometry (EDS) was obtained with an ULTRA 55 field emission scanning electron microscope (Göttingen, Germany).

The surface and cross-section of the coatings were observed by field emission-scanning electron microscopy (FE-SEM, Zeiss, ultra55, Göttingen, Germany); the working voltage was 5 keV; and the elemental composition of the coating was analyzed by energy scattering spectra (EDS, Oxford Instruments, X-max); the test voltage was 20 keV. The crystal phase was analyzed by X-ray diffraction (XRD, D/MAX-RB, Rigaku Cu K$\alpha$ radiation, $\lambda$ = 0.154056 nm). The working mode is continuous scanning, the target material is Cu, and the working parameters are as follows: scanning range 5°–90°, step size 0.1°, voltage 40 kV, and current 40 mA. The crystallinities after laser re-melting with different powers were calculated by MDI-JADE 6.0 JADE software (San Jose, CA, USA).

The corrosion resistance of the coating was evaluated by electrochemical polarization measurements in a 3.5 wt.% NaCl solution (ASTM G31-72 Standard). An EG&G Princeton Applied Research PARSTA T 2273 potentiostat with EG&G PowerSuite software (Princeton Applied Research, Oak, TN, USA) with a conventional three-electrode system was used to conduct the electrochemical test. A saturated calomel electrode with a salt bridge was used as the reference electrode, and a platinum electrode was used as the counter electrode. The specific parameters were set as follows: the scanning speed was 5 mV/min, and the measurement range was −250 mV to 500 mV (vs. OCP). In the process of the experiment, the surface of the conductor and the sample not involved in the test were sealed with epoxy resin, leaving a 10 mm × 10 mm exposed area for the electrochemical test. The measurement was started when the corrosion current was stable (amplitude of fluctuation within ± 5 mV in 5 min).

Marine atmosphere exposure experiments were carried out at one of the test stations. The sampling period is 0 a ("a" for year or years), 0.6 a, 1 a, 2 a, and 5 a. The samples were placed on the specimen holder at an angle of 45 degrees to the seaside. A digital camera was used to record the corrosion morphologies. The corrosion products were analyzed by SEM. The weight loss was calculated by removing the corrosion products.

### 2.3. Verification of the Protection Performance

The verification experiment was carried out on the research vessel Yongle; the coatings were sprayed onto the hull above the waterline from 11 November 2018 to 30 December 2019 in Mawei Shipyard. The vessel Yongle was designed by the 702 Institution of the China Shipbuilding Industry Group Co., Ltd. (Wuxi, China), with a length of 63 m and a width of 25 m. The vessel is connected by two semi-submersible structures. It is used mainly for corrosion testing in the South China Sea and is now parked in the South China sea. The coating system included an organic coating on a zinc aluminum coating for sealant purposes, as shown in Table 1.

The quality of the coating was controlled by the following standards: GB-T 8923.2-2008 [24], GB-T 8642-2002 [25], GB-T 11374-2012 [26], GB-T 18570.6-2011 [27], GB/T 3505-2009 [28], and BS EN ISO 8503-5-2004 [29], which were for the surface pretreatment, spraying parameters, and performance inspections. The anticorrosion performance was inspected every year from January 2020 to December 2022 after it was moored in the South China sea.

**Table 1.** Coating system for freeboard zones above the waterline.

| Coating Types | Dry Film Thickness/μm |
|---|---|
| Metal composite coating | 150 |
| 725-H44-61 modified high-build epoxy antirust paint (transition layer) | 50 |
| 725-H44-61 modified high-build epoxy antirust paint | 125 |
| 725-H44-61 modified high-build epoxy antirust paint | 125 |
| 725 S52-60 aliphatic polyurethane paint | 50 |
| 725 S52-60 aliphatic polyurethane paint | 50 |
| The total film thickness | 550 |

## 3. Results

### 3.1. Micro- and Macromorphology of the Coatings

Figure 1 shows the macromorphology of the arc-sprayed Zn, Zn15Al, and Zn45Al coatings. The color of the coatings turns darker as the zinc content increases. Usually, an aluminum coating has a silver metal color, and a zinc coating has a cyan color. A zinc aluminum alloy coating has a mixture of these two colors. The morphology of the coatings is uniform and coarse. The average thickness of the coating is 256, 267 μm, and 256 μm for Zn, Zn15Al, and Zn45Al coatings, respectively, and is listed in Table 2. The roughness of the coatings is $15.3 \pm 0.2$ μm, $17.3 \pm 0.2$ μm, and $12.6 \pm 0.1$ μm for Zn, Zn15Al, and Zn45Al coatings, respectively. Pores cannot be seen by the naked eye, but pores exist in each coating, which can provide ideal conditions for sealant. Theoretically, there is a synergic effect between a metal coating and an organic coating. The pores of the metal coatings provide pores for the organic anchoring location, which can increase the bonding strength between an organic coating and a substrate. The pores are also beneficial for the degradation of the organic coating via a shielding effect in the pores. The organic coating has a barrier effect, which can hold back the diffusion of the corrosion medium from reaching the substrate by blocking the holes and increasing the diffusion path length. Subsequently, the corrosion of the metal decreases. These complicated mechanisms can make the sealant metal coating have a 1.5 to 2 times longer protection life compared with the sum of organic and metal coatings independently.

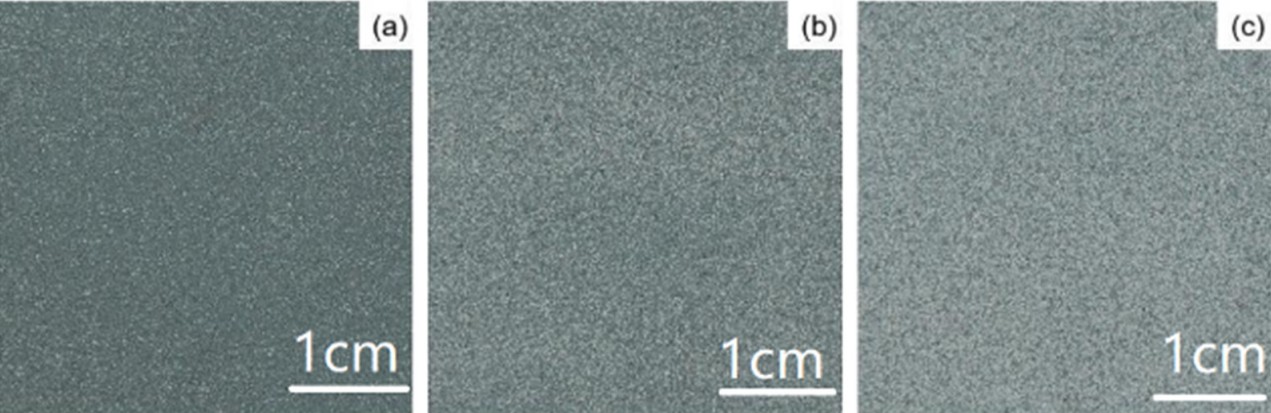

**Figure 1.** The macromorphology of the arc-sprayed zinc aluminum alloy coatings: (**a**) Zn, (**b**) Zn15Al, and (**c**) Zn45Al.

**Table 2.** Coating thickness test results for samples (μm).

| Selected Benchmark Plane | 1 | 2 | 3 | 4 | 5 | The Average |
|---|---|---|---|---|---|---|
| 1 | 177 | 249 | 186 | 244 | 301 | 231.4 ± 50.8 |
| 2 | 229 | 205 | 305 | 366 | 354 | 291.8 ± 72.5 |
| 3 | 185 | 279 | 235 | 329 | 285 | 262.6 ± 54.6 |

Figure 2 shows the micromorphology of the arc-sprayed Zn, Zn15Al, and Zn45Al coatings. It can be seen from the surface that the three kinds of coatings have the same characteristics: they are piled up with a large amount of splash flattened particles. Some small balls are embedded in and adhered to the surface. This happens during the spraying process; the melting metal is atomized by nitrogen gas. As the particles impact the substrate with a typical speed of over 80 m/s, deformation and crash occur to these melting or semi-melting particles. A flattened layer forms on the substrate. Some particles experienced a more severe crash, and subsequently, a greater number of small balls were produced during this impact. Some small particles were produced during the atomization process since it can be quickly cooled down because of its very small size, and then it retains its spherical shape. The porosities of the three coatings are listed in Table 3. All coatings have a very high porosity, ranging from 6.16% to 13.34%. Additionally, there is also a layer of pores in all coatings from 1 to 5 within a 1 cm$^2$ area. During the thermal spray process, the particles are not fully melted, and the speed of the particles is a bit low, so the particle(s) cannot be flattened completely, which leads to a very high porosity. Fortunately, this high porosity has very little effect on the anti-corrosion performance, which will be explained in greater detail in a later section.

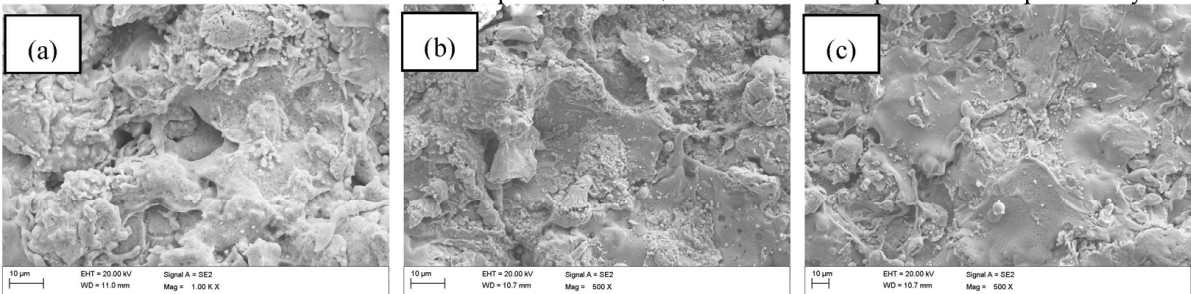

**Figure 2.** The microstructure of the arc-sprayed zinc aluminum alloy coatings: (**a**) Zn, (**b**) Zn15Al, and (**c**) Zn45Al.

**Table 3.** The three kinds of coating porosity test results (piece/cm$^2$).

| Coating Type | Sample Number | Porosity (%) | Number of Penetrated Pores |
|---|---|---|---|
| Zn | 1 | 10.16 | 2 |
| | 2 | 7.87 | 1 |
| | 3 | 13.12 | 5 |
| Zn15Al | 1 | 11.45 | 2 |
| | 2 | 10.08 | 2 |
| | 3 | 9.33 | 2 |
| Zn45Al | 1 | 8.83 | 2 |
| | 2 | 6.16 | 2 |
| | 3 | 13.34 | 4 |

*3.2. Performance Variation during Long-Term Exposure in Marine Environments*

Figure 3 shows the variation in corrosion morphologies under long-term exposure in marine environments. For all coatings, no red rust percolated from the substrate during a 5 year exposure in marine environments. No bubbles, cracks, or peel-off phenomena can be found for all coating samples. White corrosion products can be found on all samples,

which became denser and thicker as the exposure time increased. On some samples, bird excrement can be found covering part of the area of the sample, which can be dirt, and the white corrosion products are thicker. The failure of coatings is a complicated process. For sacrificial anodic metal coatings, barrier, passivation, and sacrificial effect are all functions of the coatings. Bubbles can sometimes deteriorate the electrical connection between the coatings and the substrate, which can hold back the cathodic protection effect. Only when the coating peeled off the substrate did this failure occur.

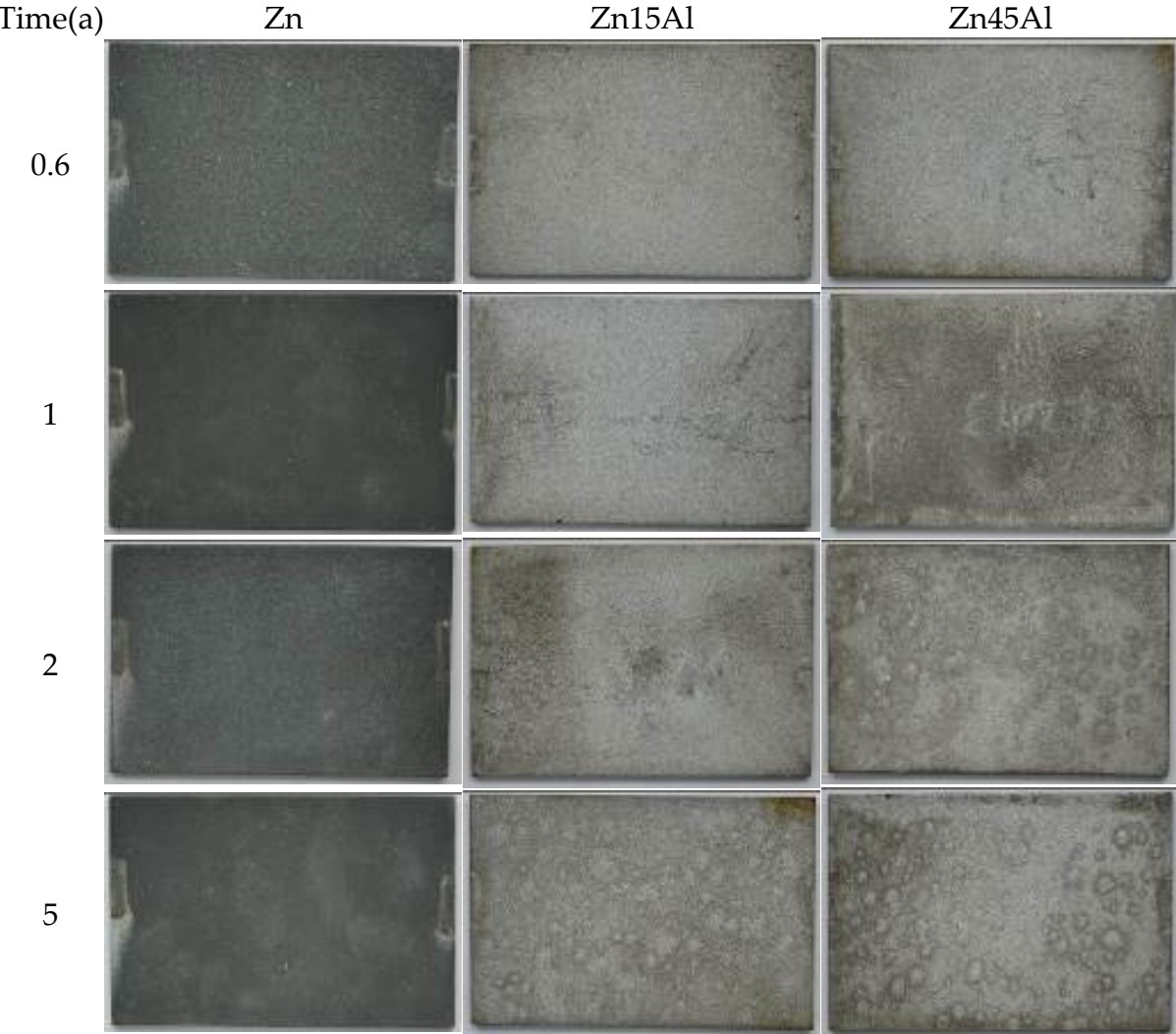

**Figure 3.** Corrosion morphology changes of the three kinds of zinc aluminum coatings in the marine atmospheric environment exposure experiment.

Figure 4 shows the bonding strength variation of the coatings during exposure to the marine environment. It can be seen that the bonding strength decreases from 8.67 to 17.85 MPa to 4.91–8.07 MPa. The Zn15Al coatings retained a bonding strength greater than 6.38 during the test, which is the highest value among the three kinds of coatings. Bonding strength is a key property of coatings that resist peeling off. Usually, corrosion products induce stress both in the coatings and between the interfaces, which could lead to the coating peeling off and decreasing its bonding strength. It is useful to prevent the coating from peeling off by decreasing the corrosion rate of the coating. It can also be found that zinc alloy coatings have a slower corrosion rate than zinc coatings. During the exposure period, the corrosion medium diffused into the coating and reached the substrate. First, corrosion happened to the coatings, and the corrosion products have a much lower bonding

strength compared to the metal. Additionally, the corrosion products have a bigger volume compared to metal, and this volume increase by the corrosion products can decrease the bonding strength and even spall the coatings. The decrease in bonding strength, or peeling off, is the main failure type of metal coatings.

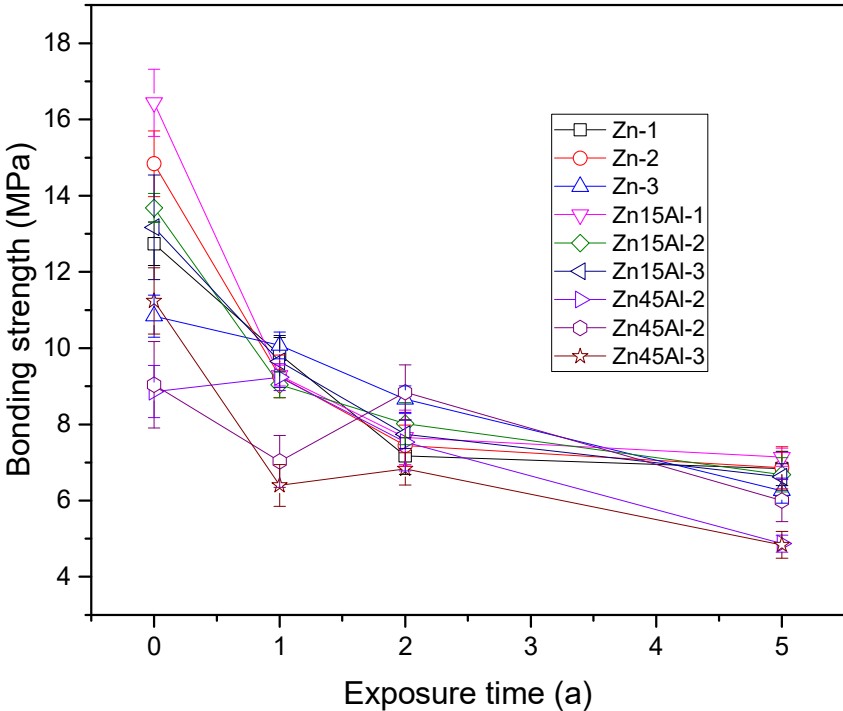

**Figure 4.** Bonding strength variations with time of the arc-sprayed zinc aluminum alloy coatings.

Figure 5 shows the microcorrosion morphologies variation during long-term exposure in marine environments. The three kinds of corrosion products all have a very dense structure, which can protect the coating from further corrosion. The corrosion products can also block the pores of the coatings. Figure 6 shows the weight loss of the coating after 5 years of exposure. It can be seen that the zinc coating has a very high corrosion rate, ranging from 118 μm/a in the first half year to about 40–50 μm/a during the remainder of the time. The Zn15Al weight loss remains more stable, and for some years, a weight gain was found. Based on these experimental results, the use of zinc and its alloy coatings is specified. A zinc coating cannot be used in water or in marine atmosphere environments because it has a very high corrosion rate. Whereas Zn15Al and Zn45Al can be used independently because they have a much lower corrosion rate even without a sealant.

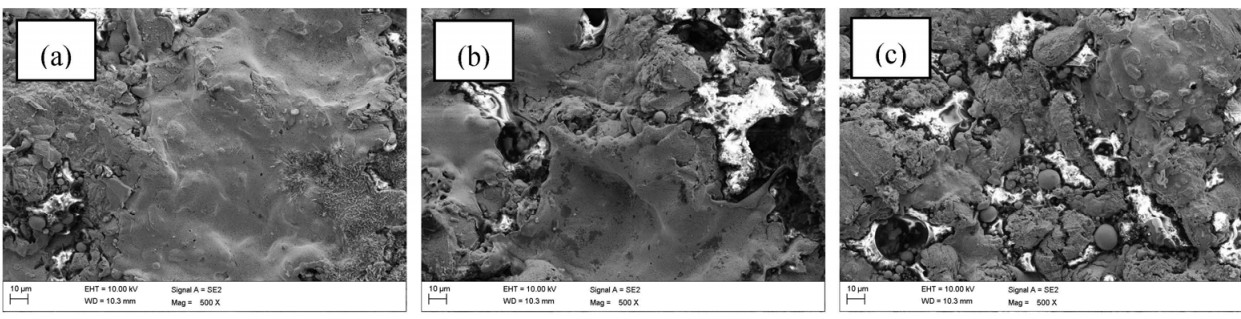

**Figure 5.** Corrosion morphologies of the arc-sprayed zinc aluminum coatings: (**a**) Zn, (**b**) Zn15Al, and (**c**) Zn45Al.

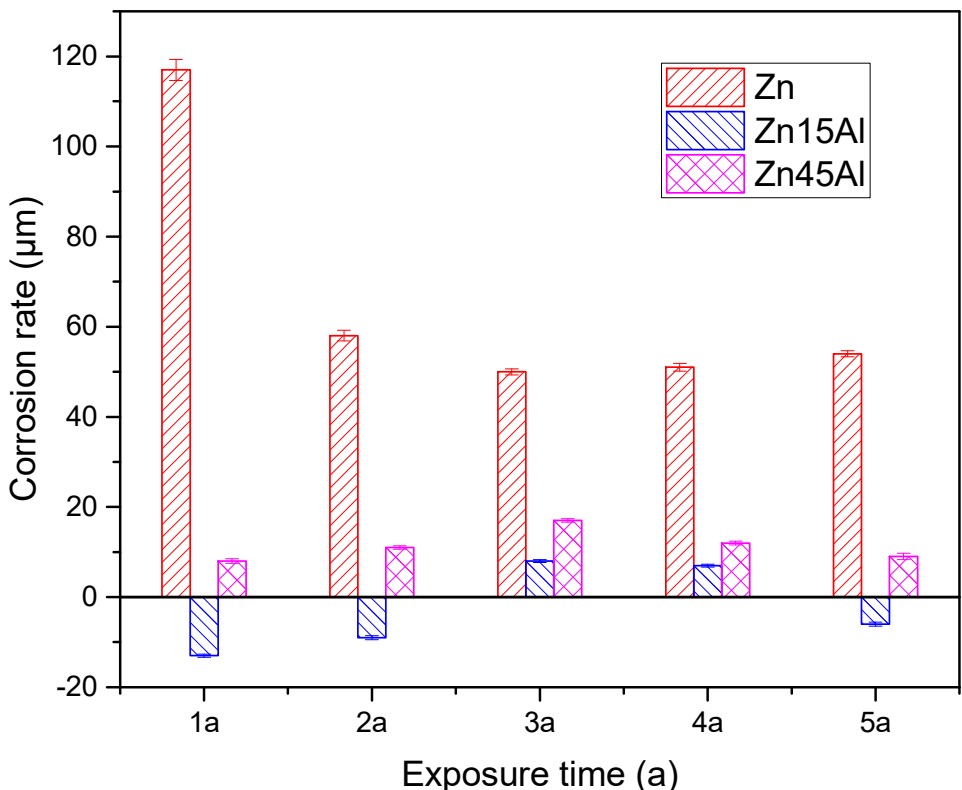

**Figure 6.** Time−dependent weight loss of the arc−sprayed zinc aluminum coatings during the exposure experiment.

The exposure periods contain one distinct half-semi-circle loop in the Nyquist plots at a higher studied frequency and a small tail at a lower studied frequency (Figures 6 and 7). The lower frequency plots are attributed to the deposition of corrosion products on the surface after the reaction of the coating with the solution at the coating/solution interface. This can be protective and contribute to reducing the active surface area of the coating. Furthermore, the thickening of corrosion products simultaneously occurs when the coating begins to dissolve after exposure to the solution. Initially, the corrosion products can be porous/defective and thereby allow the ingress of aggressive ions from the solution toward the coating surface and result in the formation of corrosion products. When the duration of exposure periods increases, the thickness of corrosion products increases, and thus the dimensions of semi-circle loops increase and improve the corrosion resistance. The coating contains Al and Zn, which are active metals for dissolution, and begins to corrode after exposure to the $Cl^-$ solution.

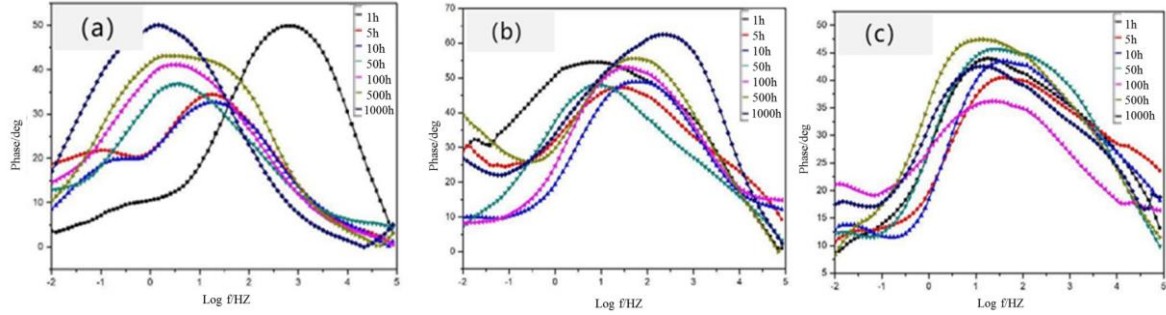

**Figure 7.** The Bode plot variation of the arc-sprayed zinc aluminum coatings during the exposure experiment: (**a**) Zn, (**b**) Zn15Al, and (**c**) Zn45Al.

Figure 7 shows the Bode plot of the coating in seawater. At the first stage of immersion, there is only one time constant in the Bode plot. As the immersion time increases and corrosion products accumulate on the surface of each coating, there are two constants in the Bode plot. Figure 8 shows the corresponding equivalent circuit of the coating structure in seawater as exposure time increased, during which the circuit components are described as $R_s$, $R_p$, $R_{ct}$, $R_c$, $W$, and $Q$ (*CPE*) that correspond to solution resistance, polarization resistance, charge transfer resistance, corrosion product film resistance, Warburg constant, and constant phase elements, respectively. The $R_p$ variation of the coating samples during the exposure test is shown in Figure 9. It can be seen that the $R_p$ of the coating increased at first and then decreased as the exposure time increased. As the formation of the corrosion products increases, $R_p$ increases. As the exposure time increased, the corrosion products reached their maximum thickness. After that, the corrosion of the coating was the dominant process, and $R_p$ decreased as the exposure time increased. It can be seen from the Bode plot that at the first stage, there is only one time constant (active electrochemical process), which means that the coatings are in an active solving process. As the exposure time increases, a much thicker and denser layer of corrosion products forms on the surface, so that there are two time−constants (an active electrochemical process and a diffusion of electrolyte).

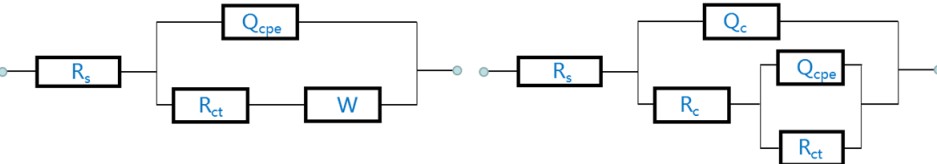

**Figure 8.** The equivalent circuit of the arc-sprayed zinc aluminum coatings during the exposure experiment according to the spectra in Figure 7a RQ(RW) and Figure 7b R(Q(R(QR))).

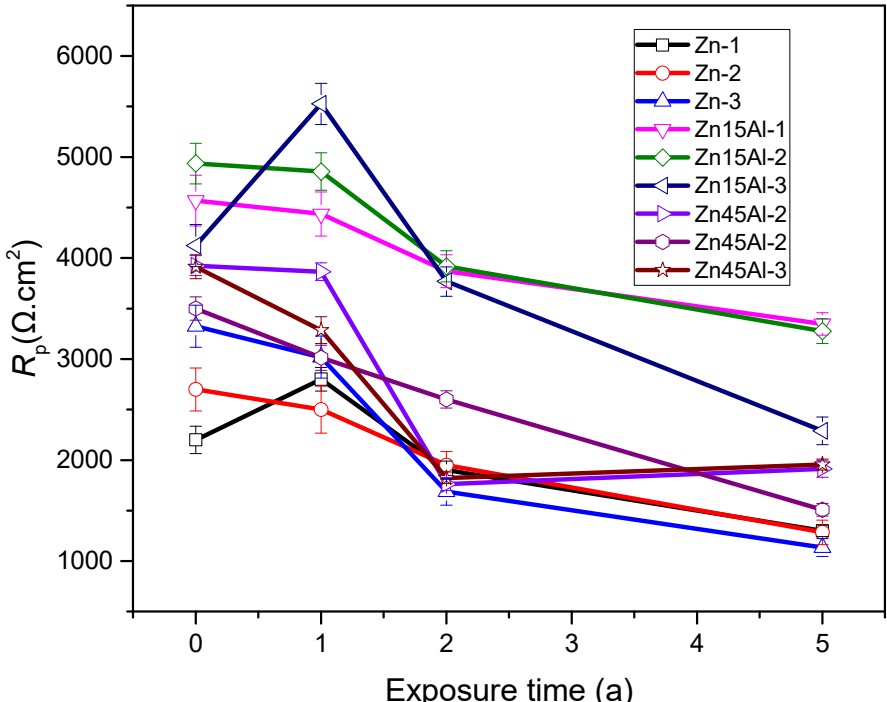

**Figure 9.** Changes in the polarization resistance of the arc-sprayed zinc aluminum coatings during the exposure experiment according to the spectra in Figure 7.

The numerical value for all the coatings is in the same order of magnitude, which is 1500–550 $\Omega \cdot cm^2$. The Zn15Al coating has a more stable $R_p$ than that of Zn and Zn45Al. The $R_p$ value of the Zn15Al coating remains greater than 2200 $\Omega \cdot cm^2$. It means that the

Zn15Al coating has a slower corrosion rate than the other two coatings. For zinc and its alloy coatings, a sacrificial effect and a barrier effect are both anticipated when they protect the steel substrate from corrosion. However, the coating cannot endure for a long period if the sacrificial effect is too high. Ideally, it should be maintained at a suitable level.

The variation in corrosion potential with time for the three kinds of coatings is listed in Table 4. It can be seen that the corrosion potentials of all the coatings are negative compared to the Q235 steel samples. The value is from −0.787 to −0.943 V (vs. SCE), which can provide enough cathodic protection for steel throughout the experimental exposure time. The potential is about 100 mV more positive than pure zinc metal, which means that the coatings cannot separate the steel substrate from the corrosion medium; the steel substrate also takes part in the corrosion process. However, it contributes little current, and as a result the corrosion process is mainly determined by the corrosion process of the zinc and its alloy coatings.

**Table 4.** Changes in corrosion potential over time for the three kinds of arc-sprayed zinc aluminum coatings ($E/V_{\text{vs. OCP}}$).

| Time (d) | Zn | Zn15Al | Zn45Al | Q235 Steel |
|---|---|---|---|---|
| 1 | −0.787 | −0.819 | −0.832 | −0.685 |
| 2 | −0.806 | −0.836 | −0.854 | −0.667 |
| 5 | −0.847 | −0.943 | −0.835 | −0.623 |
| 10 | −0.819 | −0.915 | −0.922 | −0.634 |
| 20 | −0.807 | −0.897 | −0.866 | −0.629 |
| 50 | −0.833 | −0.886 | −0.856 | −0.651 |
| 100 | −0.825 | −0.878 | −0.835 | −0.638 |
| 200 | −0.842 | −0.861 | −0.876 | −0.665 |
| 500 | −0.837 | −0.884 | −0.853 | −0.627 |
| 1000 | −0.843 | −0.856 | −0.879 | −0.631 |

*3.3. Verification Experiment*

Figure 10 shows the typical morphologies of the vessel Yongle after sandblasting a bulk area and a welding area. It is required that the surface reaches a level of Sa 2.5 (the level of the roughness and clearance), that no pollutant or oil is on the surface, that the roughness is larger than 65 μm, and that there is no embedment of sand in the steel substrate to assure a high-quality bonding. The other parameters were also tested before spraying, as listed in Table 5, including humidity, temperature, and salinity. All parameters meet the requirements according to the standard adopted in this experiment.

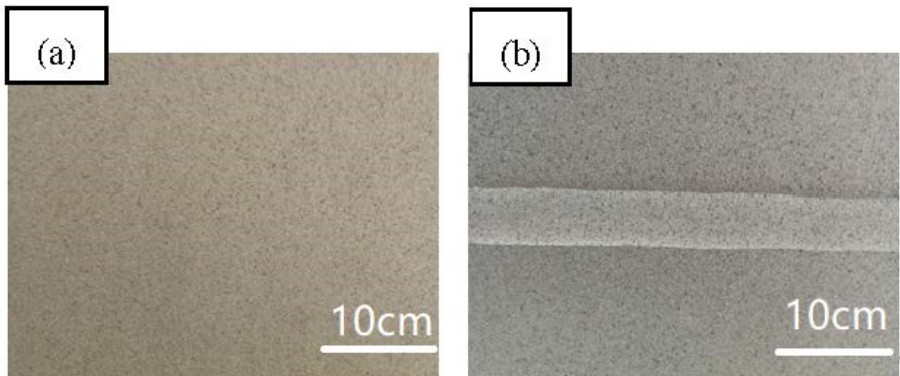

**Figure 10.** The typical surface status of the ship's hull after sandblasting: (**a**) bulk area and (**b**) welding area.

**Table 5.** The typical surface pretreatment qualities and environmental parameters of the ship's hull after sandblasting and before spraying.

| Parameters | Relative Humidity (%) | Ship's Hull Temperature (°C) | Atmospheric Temperature (°C) | Dew Point (°C) | Temperature Difference (°C) |
|---|---|---|---|---|---|
| Testing data | 62.5 | 19.9 | 20.0 | 13.0 | 6.9 |
| Parameters | Roughness (Rz/μm) 1 | | | Roughness (Rz/μm) 2 | |
| Testing data | 153.2 | | | 176.6 | |
| Parameters | Salinity (mg/m$^2$) 1 | | | Salinity (mg/m$^2$) 2 | |
| Testing data | 13.92 | | | 15.66 | |

To avoid the influence of fog and dew, spraying was usually carried out during the daytime. After spraying each small part, the coating quality was tested by examining the surface appearance, bonding strength, and thickness. Typical data are listed in Tables 6 and 7. The bonding strength is 14.14–18.68 MPa, higher than the contract requirement of 7 MPa according to the standard NORSOK M-501-2012 Coating System Guide [30]. The whole sprayed surface area is about 3700 m$^2$ around the vessel hull above the waterline.

**Table 6.** Coating thickness test results for the ship hull (μm).

| Test Site | 1 | 2 | 3 | 4 | 5 | Average |
|---|---|---|---|---|---|---|
| 1 | 333 | 300 | 310 | 324 | 375 | 328.4 ± 29.0 |
| 2 | 296 | 476 | 318 | 365 | 523 | 395.6 ± 99.4 |
| 3 | 199 | 179 | 256 | 247 | 256 | 227.4 ± 35.9 |
| 4 | 265 | 297 | 269 | 292 | 270 | 278.6 ± 14.7 |
| 5 | 231 | 200 | 224 | 216 | 259 | 226.0 ± 21.8 |
| 6 | 195 | 162 | 260 | 154 | 214 | 197.0 ± 42.8 |
| 7 | 195 | 307 | 285 | 163 | 189 | 227.8 ± 63.9 |
| 8 | 346 | 245 | 393 | 289 | 352 | 325.0 ± 58.1 |
| 9 | 322 | 403 | 366 | 339 | 378 | 361.6 ± 31.9 |
| 10 | 205 | 192 | 199 | 151 | 180 | 185.4 ± 21.4 |

**Table 7.** Bonding strength test results (schedule).

| Serial Number | Structure/Location | Bonding Strength/MPa | Thickness/μm |
|---|---|---|---|
| 1 | Wet deck center 1 | 14.64 | 233 |
| 2 | Wet deck center 2 | 14.14 | 256 |
| 3 | Wet deck center 3 | 18.68 | 287 |

The corrosion morphology of the coating was examined every year starting in 2020. The morphology of the vessel hull after 3 years is shown in Figure 11. It can be seen from the figure that there is not a single corrosion spot on the ship hull above the waterline. There is also no peeling off, cracking, or bubbles on the coatings. Only some marine life can be found around the waterline.

High porosity is a typical characteristic of arc spray coating, and penetrated pores cannot be eliminated since the speed of the particle during spray is relatively low, which cannot be avoided. Fortunately, the high porosity does influence the anticorrosion performance of the arc-sprayed zinc and its alloy coating significantly. The protection mechanisms of the zinc and its alloy metal coating include the barrier effect, a sacrificial anodic effect, and a passivation effect. Corrosion products can block the pores effectively to hold back the diffusion of the corrosion medium, which can prevent corrosion from occurring. Additionally, the zinc and its alloy coating are usually used with an organic sealant coating.

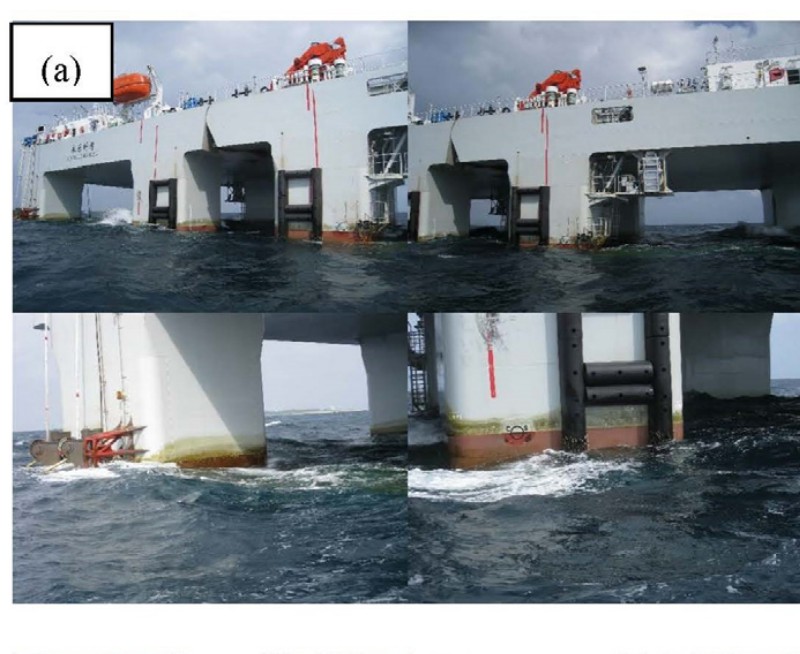

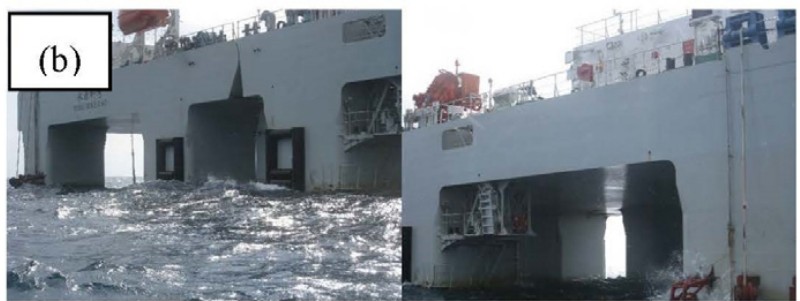

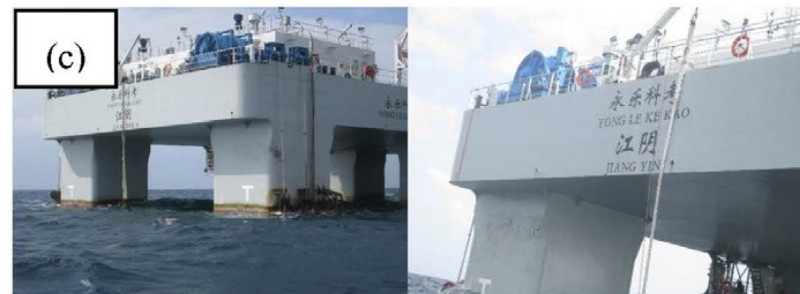

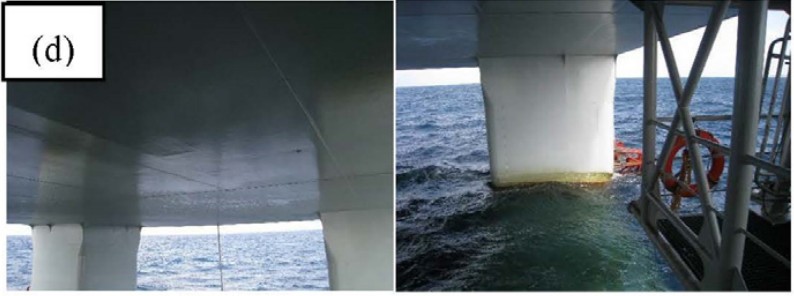

**Figure 11.** Corrosion status of the ship's hull at the freeboard zones of the vessel Yongle after 3 years in the South China Sea. (**a**) Overall appearance. (**b**) Port side of the platform. (**c**) The starboard side of the platform. (**d**) The bottom of the platform.

Theoretically, there is a synergic effect between the metal coating and an organic coating. The pores of the metal coatings provide pores for the organic anchoring location, which can increase the bonding strength between the organic coating and the substrate. The pores are also beneficial for the degradation of the organic coating via a shielding effect in the pore. The organic coating has a barrier effect, which can hold back the diffusion of corrosion medium from reaching the substrate by blocking the holes and increasing the diffusion path length. Subsequently, the corrosion of the metal decreases. These complicated mechanisms can make the sealant metal coating have a 1.5 to 2 times longer protection life compared with the sum of the organic and metal coatings.

In this paper, the field exposure experiment shows that zinc alloy coatings have a slower decrease rate than zinc coatings. During the exposure period, the corrosion medium diffused into the coating and reached the substrate. First, corrosion occurred to the coatings; the corrosion products have a much lower bonding strength compared to the metal. Additionally, the corrosion products have a bigger volume than the metal, which can decrease the bonding strength and even cause spalling of the coatings. The decrease in bonding strength, or peeling off, is the main failure type for coatings. In this paper, the coatings will not experience peeling off for a very long period since few corrosion products are accumulated in the coatings.

The corrosion rates of the coatings indicate that only a zinc coating has a very high corrosion rate, while Zn15Al and Zn45Al coatings both have a very low corrosion rate, which means the coating could not be depleted in at least 50 years in marine atmospheric environments.

As mentioned above, the zinc aluminum alloy coating with an organic sealant will not be peeled off and depleted in the South China Sea marine atmosphere environment, and it can provide reliable long-term protection for steel structures in the South China Sea environment.

## 4. Conclusions

In this paper, three kinds of arc-sprayed zinc and its alloy coatings were evaluated to determine their long-term protection performances by examining the time-dependent variation of Rp and bonding strength. The following conclusions can be drawn:

1.  Arc sprayed Zn, Zn15Al, and Zn45Al can provide efficient corrosion protection for steel in the South China Sea marine environments; Zn15Al has the lowest corrosion rate and the best protection effect;
2.  The bonding strength and the $R_p$ will decrease as time increases, which can lead to performance degradation, but 5 years of exposure indicates that the degradation is acceptable for long-term protection;
3.  The arc-sprayed Zn15Al coating, together with the organic sealant, exhibits excellent corrosion protection for the ship's hull above the waterline for 3 years; no single corrosion spot is found on the whole Yongle vessel;
4.  The verification experiments indicate that arc-sprayed zinc and its alloy coatings can provide efficient long-term protection for steel structures in atmospheric environments, although the corrosion conditions are very harsh.

**Author Contributions:** Conceptualization, L.M.; methodology, Z.-L.L. and Y.-L.X.; investigation, X.-S.Z., G.-S.H. and M.-X.S.; writing—original draft preparation, X.-B.L.; writing—review and editing, G.-S.H. All authors have read and agreed to the published version of the manuscript.

**Funding:** This research was funded by the foundation of the state-key laboratory of solid lubrication, Lanzhou Institute of Chemical Physics, Chinese Academy of Science, grant number SLS 2018.

**Institutional Review Board Statement:** Not applicable.

**Informed Consent Statement:** Not applicable.

**Data Availability Statement:** Not applicable.

**Acknowledgments:** The author thanks Xudong Sui for providing anti-abrasive test conditions. The author thanks Qingjun Chen for providing metal wires.

**Conflicts of Interest:** The authors declare no conflict of interest.

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
