# Peer review of "Degradation Behavior of Arc-Sprayed Zinc Aluminum Alloy Coatings for the Vessel Yongle in the South China Sea"

_coatings, doi:10.3390/coatings13071139_

Round 1
Reviewer 1 Report
Dear authors,
the article is well written and it gives important information about the coatings durability in harsh conditions. Please just improve the introduction. Addition of more international bibliography is highly recommended.
Author Response
The response is listed in the attachment.

Reviewer 2 Report
Attached

Needs lots of work.
Reviewer 3 Report
General description
The article concerns degradation behavior of arc sprayed zinc aluminum alloy coating for Yongle vessel in the South China Sea.
The article compares the results of corrosion tests of 3 anticorrosion coatings.
The Introduction lacks the purposefulness of the conducted research and a broader discussion of the state of knowledge. The background review is based on only 15 items.
The research methods and the analysis of the results have been described, but require supplementation.
Detailed comments
1. The Introduction is too general. Please expand it.
2. Please highlight in the Introduction what new your research introduces.
3. Materials and Methods: There is no information on what device was used to make the coatings. Please complete this information.
4. Line 41: No information on the chemical composition of Q235 steel.
5. Line 78: In the text of the article there is information that SEM tests were carried out on samples that were in a salt spray chamber for 5 days. In line 169, however, there is information that Figure 5 shows morphologies during long-term exposure in marine environments. What samples are these images in Figure 5 from?
6. Line 81: All symbols used in the article should be explained. This makes it easier for the reader to analyze the content. Please add information on what "a" means.
7. Results: How was the coating thickness determined? On what device were these tests performed?
8. Line 102: It should be Table 2 not Table 1.
9. Figure 1: The photos in Figure 1 do not add much to the article. It would be better to show the structure of the coating on a cross section.
10. Line 129: Instead of Table 2 should be Table 3.
11. Line 130: The porosity in Table 3 changes from 6.16% (not 6.18%).
12. Figures 2, 5, 10 and 11: Lack the designations of "a", "b", ... In the captions under the figures there are references to these designations.
13. Descriptions in the drawings should be in English. Please translate the descriptions in Figures 4 and 9 and Table 4.
14. Figure 4: The drawing is illegible.
15. Lines 170 - 173: It is not possible to tell from Figure 5 what the corrosion products are. Zinc and zinc aluminium coatings containing zinc (zinc corrodes preferentially in relation to aluminum) in an environment containing chlorides creates corrosion products containing chlorine. The most likely corrosion product of zinc and zinc aluminium coatings is simonkolleite Zn5(OH)8Cl2∙H2O. Authors wishing to explain the mechanism of the occurring reactions should study corrosion products or carry out an analysis based on the existing literature.
16. Figure 6: The sentence starts with a lowercase letter.
17. Figure 6: The authors in Figure 6 show the weight loss of the coatings (no axis description in the graph). The Zn15Al coating shows mass loss in 3 cases. Please verify and explain what Figure 6 shows.
In addition, the weight change values ​​should have measurement errors marked.
18. Figure 7: Please explain what Figure 7 shows.
19. Figure 7: The description of figure lacks the descriptions of figs. a-c.
20. Line 218: It should be -0.943 V, not -0.922 V.
21. Table 4: Units are missing from the table.
12. Line 228: All symbols used in the article should be explained. Please add information on what “Sa” stands for.
23. Line 242: There is no information that the text refers to Table 7.
24. Line 242: The text says that the expected value is 7 MPa. There is no reference to the standard.
25. Tables 6: Average values ​​should contain measurement error information.
26. Figure 13: The drawings is described differently. Please standardize the way of describing drawings.
27. Figure 13: It would be useful to enlarge the fragments above the waterline to confirm the conclusions of line 254.
28. Please explain the discrepancy between conclusion 4 and the text on line 179.
Round 2
Reviewer 1 Report
Dear authors,
thanks for providing the reviewed manuscript. In my opinion it is suitable for publication
Author Response
Thank you very much for your approve on our manuscript.
Reviewer 2 Report
Important notes
The authors did a good job revising the manuscript. However, the following point still need revising with great attention.
1. Some sentences are duplicated in the abstract and the introduction. Authors should solve this issue. Mostly, the tenses that are used in the abstract and introduction must be past perfect and present perfect.
2. Line 65: Further more, zinc and its ….. → Furthermore, zinc and its ………. , also, Line 65 rapidly [23 → rapidly [23]
Line 92: The bonding strength (BS) of the coatings were collected by an …..
→ The bonding strength (BS) of the coatings was collected by an …..
3. The article still needs proofreading, especially the abstract and the introduction. It is a vital necessity.
4. We still believe that figures 11 and 12 are not needed. Delete.
Still needs attention in many places.
Author Response
Thank you very much for your valuable time. The responses are enclosed in the attachment.

Reviewer 3 Report
The article concerns degradation behavior of arc sprayed zinc aluminum alloy coating for Yongle vessel in the South China Sea.
The article compares the results of corrosion tests of 3 anticorrosion coatings.
1. The Introduction lacks the purposefulness of the conducted research and a broader discussion of the state of knowledge. The background review is based on only 15 items.
Thank you very much for your positive comment on our manuscript, we revised the introduction by extending the literatures, by discussing more on the problems need to be solved in this area.
Reviewer: The authors have extended the literature review.
2. The research methods and the analysis of the results have been described, but required supplementation.
Methods and equipment were supplemented, including the spray equipment and other test equipments.
Reviewer: The authors corrected the article in accordance with the comments, with the exception of comment no. 15.
Comment to answer no. 4 contains my suggestion.
Detailed comments
1. The Introduction is too general. Please expand it.
Thank you very much for your positive comment on our manuscript, we revised the introduction by extending the literatures, by discussing more on the problems need to be solved in this area.
Reviewer: The authors have extended the literature review.
2. Please highlight in the Introduction what new your research introduces.
The bonding strength degradation behavior is taken into consideration in this paper. And it has been added into the introduction.
Reviewer: The authors have completed the Introduction.
3. Materials and Methods: There is no information on what device was used to make the coatings. Please complete this information.
The device was added in the text.
Reviewer: The authors supplemented the article in this regard.
4. Line 41: No information on the chemical composition of Q235 steel.
The nominal composite from the manufacturer was listed in the text.
Reviewer: The authors added information on the chemical composition of Q235 steel in the article. Please add "wt." before the "%" symbol (percentage by weight).
5. Line 78: In the text of the article there is information that SEM tests were carried out on samples that were in a salt spray chamber for 5 days. In line 169, however, there is information that Figure 5 shows morphologies during long-term exposure in marine environments. What samples are these images in Figure 5 from?
Sorry for misleading, the salt spray test part was delete in the text, since there is no salt spray test carried out in the present work.
Reviewer: The text has been removed.
6. Line 81: All symbols used in the article should be explained. This makes it easier for the reader to analyze the content. Please add information on what "a" means.
All symbols were checked and explained in the text. Here “a” means a year, a standard expression for year/years in corrosion field.
Reviewer: The authors provided information on the designation.
7. Results: How was the coating thickness determined? On what device were these tests performed?
For samples, the thickness was measured randomly on the surface at five points, the average thickness was calculated for each sample. For ship hull during the coating construction, the thickness was measured regularly on a plane surface within 10cm×10cm area, the average thickness was calculated for each sampling location.
The device is a Elcometer 456 Coating Thickness Gauge (Elcometer, England).
Reviewer: The authors extended their article in this regard.
8. Line 102: It should be Table 2 not Table 1.
Revised, it is Table 2.
Reviewer: The authors made a correction.
9. Figure 1: The photos in Figure 1 do not add much to the article. It would be better to show the structure of the coating on a cross section.
We didn’t examine the cross section of all samples. There is no enough time for us to do the experiment before the deadline. Thank you for your understanding.
Reviewer: The authors explain the reason for not posting photos. On my part, changing the photos was a suggestion.
10. Line 129: Instead of Table 2 should be Table 3.
Revised, it is Table 3.
Reviewer: The authors made a correction.
11. Line 130: The porosity in Table 3 changes from 6.16% (not 6.18%).
Revised.
Reviewer: The authors made a correction.
12. Figures 2, 5, 10 and 11: Lack the designations of "a", "b", ... In the captions under the figures there are references to these designations.
Designations are added for Figures 2, 5, 10 and 11.
Reviewer: I have no comments to the method of marking.
13. Descriptions in the drawings should be in English. Please translate the descriptions in Figures 4 and 9 and Table 4.
The descriptions in Figures 4 and 9 and Table 4 were translated to English.
Reviewer: The authors made a correction.
14. Figure 4: The drawing is illegible.
Figure 4 was re-drawn with Origin software.
Reviewer: The drawing is more readable.
15. Lines 170 - 173: It is not possible to tell from Figure 5 what the corrosion products are. Zinc and zinc aluminium coatings containing zinc (zinc corrodes preferentially in relation to aluminum) in an environment containing chlorides creates corrosion products containing chlorine. The most likely corrosion product of zinc and zinc aluminium coatings is simonkolleite Zn5(OH)8Cl2∙H2O. Authors wishing to explain the mechanism of the occurring reactions should study corrosion products or carry out an analysis based on the existing literature.
The corrosion products for zinc and its alloys are complicated and environmental sensitive. We didn’t do much test on these aspects. Wen only get the data from the XRD, which is not posted here.
Reviewer: In my opinion, the authors did not improve their work in this area. The work should include an analysis of corrosion products based on test results or literature.
Authors may also remove from the article information on corrosion products not confirmed by research results.
16. Figure 6: The sentence starts with a lowercase letter.
Revised.
Reviewer: The authors have changed the caption under the drawing. I do not have any comments.
17. Figure 6: The authors in Figure 6 show the weight loss of the coatings (no axis description in the graph). The Zn15Al coating shows mass loss in 3 cases. Please verify and explain what Figure 6 shows.
In addition, the weight change values ​​should have measurement errors marked.
Axis descriptions are added in the graph. The error bars were also added.
Reviewer: The authors supplemented the axis descriptions.
18. Figure 7: Please explain what Figure 7 shows.
The bode plot was explained.
Reviewer: The authors supplemented the article in this regard.
19. Figure 7: The description of figure lacks the descriptions of figs. a-c.
The descriptions of figs. a-c were added.
Reviewer: I do not have comments on the way of marking drawings.
20. Line 218: It should be -0.943 V, not -0.922 V.
Revised.
Reviewer: I do not have any comments.
21. Table 4: Units are missing from the table.
Added in the title and the table.
Reviewer: The authors supplemented the table.
12. Line 228: All symbols used in the article should be explained. Please add information on what “Sa” stands for.
“Sa” stands for the level of the roughness and clearance.
Review: The authors supplemented the text.
23. Line 242: There is no information that the text refers to Table 7.
The information was added to Table 7.
Reviewer: I do not have any comments.
24. Line 242: The text says that the expected value is 7 MPa. There is no reference to the standard.
the contract required 7 MPa according to the standard NORSOK M-501-2012 Coating System Guide
Reviewer: I do not have any comments.
25. Tables 6: Average values ​​should contain measurement error information.
The deviation was added in the Table 6.
Reviewer: The authors supplemented the table.
26. Figure 13: The drawings is described differently. Please standardize the way of describing drawings.
The drawings were described standardized.
Reviewer: The authors unified the descriptions of the drawings.
27. Figure 13: It would be useful to enlarge the fragments above the waterline to confirm the conclusions of line 254.
Thank you for your valuable comment.
We did do the check for every corrosion detail of the ship hull, but we only took the overall pictures and didn’t take the detail photos. It will take much to make an examination. We only have two sailors on the Yongle vessel, they are unable to take the photos near the waterline by themselves.
I am afraid that we could not provide enlarge the fragments above the waterline.
Reviewer: The authors explained the reason for not posting additional photos. I do not have any comments.
28. Please explain the discrepancy between conclusion 4 and the text on line 179.
Sorry for misleading by not mentioning the organic sealant. In the text, the Zn coating was also sealed to ensure a longterm protection in marine atmosphere environment.
Reviewer: I do not have any comments.
Author Response

(The authors gave the same response as above.)

Round 3
Reviewer 3 Report
I recommend this paper for publication.